# Effect of High-Flow Nasal Oxygenation on Gastric Insufflation in Patients Undergoing Laryngeal Microsurgery under Tubeless General Anesthesia with Neuromuscular Blockade

**DOI:** 10.3390/jcm12051800

**Published:** 2023-02-23

**Authors:** Min Ying Chang, Hyun Jeong Kwak, Jong Yeop Kim, Ji Young Park, Hee Yeon Park, In Kyong Yi

**Affiliations:** 1Department of Anesthesiology and Pain Medicine, Ajou University School of Medicine, Suwon 16499, Republic of Korea; 2Department of Anesthesiology and Pain Medicine, Gachon University Gil Medical Center, Incheon 21565, Republic of Korea

**Keywords:** apnea, insufflation, nasal cavity, neuromuscular blockade, oxygen, pyloric antrum

## Abstract

Background: High-flow nasal oxygenation is an oxygen delivery method by which high concentrations of heated humidified oxygen are supplied via the nasal cavity. This study aimed to investigate the effect of high-flow nasal oxygenation on gastric volume change in adult patients undergoing laryngeal microsurgery under tubeless general anesthesia with neuromuscular blockade. Methods: Patients aged 19–80 years with an American Society of Anesthesiologists physical status 1 or 2 who were scheduled to undergo laryngoscopic surgery under general anesthesia were recruited. Patients received high-flow nasal oxygenation therapy at 70 L/min during surgery under general anesthesia with neuromuscular blockade. The cross-sectional area of the gastric antrum was measured via ultrasound in the right lateral position before and after high-flow nasal oxygenation, and the gastric volume was calculated. The duration of apnea, i.e., the duration of administration of high-flow nasal oxygenation in the paralyzed state, was also recorded. Results: Of the 45 patients enrolled, 44 completed the study. There were no significant differences in the antral cross-sectional area in the right lateral position, gastric volume, and gastric volume per kg between before and after high-flow nasal oxygenation application. The median duration of apnea was 15 (interquartile range, 14–22) min. Conclusion: High-flow nasal oxygenation at 70 L/min during apnea with the mouth open did not influence the gastric volume in patients undergoing laryngeal microsurgery under tubeless general anesthesia with neuromuscular blockade.

## 1. Introduction

High-flow nasal oxygenation (HFNO) is an oxygen delivery method providing high concentrations of heated humidified oxygen at a flow rate of up to 70 L/min through the nasal cavity. It is effective in preventing atelectasis and bronchospasm by providing continuous positive airway pressure (CPAP), promoting dead space washout, increasing alveolar recruitment, and decreasing the work of breathing [1,2]. HFNO can be used to prevent hypoxemia through pre-oxygenation and apneic oxygenation during the peri-intubation period of general anesthesia induction [3,4]. In addition, by providing apneic oxygenation and ventilation, it enables tubeless anesthesia, which allows unblocked access to all parts of the glottis during laryngeal microscopic surgery [5,6].

Non-invasive positive pressure ventilation, including CPAP and bilevel positive airway pressure (PAP), can cause gastric distension and increased gastric secretion, which can subsequently lead to gastric content regurgitation and pulmonary aspiration [7]. Theoretically, HFNO can also cause gastric distension and regurgitation since it also generates CPAP. However, previous studies found no gastric content regurgitation when pre-oxygenation with HFNO was performed for rapid sequence induction in emergency surgery [8], and no gastric distension or increase in gastric secretion when HFNO was applied at flow rates of up to 70 L/min for 30 min in healthy patients with spontaneous breathing [9]. To the best of our knowledge, no studies have reported the effect of intraoperative HFNO on gastric insufflation or gastric volume change in patients under general anesthesia with neuromuscular blockade. Since the patient’s airway becomes unprotected when tubeless anesthesia with neuromuscular blockade is applied, it may be of clinical significance to evaluate the effect of HFNO on gastric insufflation.

Thus, this study aimed to evaluate the effect of HFNO on gastric volume change in patients undergoing laryngeal microscopic surgery under tubeless general anesthesia with neuromuscular blockade.

## 2. Materials and Methods

### 2.1. Patients

This single group prospective observational study was approved by the Institutional Review Board of Ajou Hospital, Suwon, South Korea, on 19 November 2020 (approval no. AJIRB-MED-DE1-20-424) and registered at ClinicalTrials.gov on 16 November 2020 (NCT04629911). All patients provided written informed consent prior to participation in the study, and procedures were conducted in accordance with the Helsinki Declaration 2013.

Patients aged 19–80 years with an American Society of Anesthesiologists physical status 1 or 2 who were scheduled to undergo laryngoscopic surgery under general anesthesia were recruited. Patients with gastroesophageal reflux, gastric pathology, pregnancy, pulmonary comorbidities (e.g., asthma, emphysema, chronic obstructive pulmonary disease, etc.), and severe obesity (body mass index > 35 kg/m^2^) were excluded.

### 2.2. Interventions

According to the fasting standards, all patients fasted for at least 2 h for clear fluids and at least 8 h for solid foods. Two anesthesiologists, who had conducted more than 50 gastric sonography examinations, independently performed ultrasound assessment of the gastric volume prior to anesthesia induction (baseline) and at the completion of HFNO. The latter assessment was performed after completion of surgery while maintaining anesthesia and HFNO therapy. All assessments were performed with a Minolta ultrasound machine (SonimageHS1, Konica Minolta, Japan) at a low frequency (2–5 MHz) using a curvilinear transducer. The values measured by the two anesthesiologists were averaged. Blinding was not applied for anesthesiologists performing ultrasound.

Ultrasound gastric volume assessment was performed according to the standardized scanning protocol [10]. Perla et al. [10] reported that the gastric antrum provided the most reliable quantitative information for gastric volume particularly when participants were in the right lateral decubitus position. Therefore, the patients were placed in the right lateral decubitus (RLD) position. To visualize the gastric antrum, the transducer was placed along the aorta at the abdominal aorta level, investigating the cross-section in the sagittal plane between the left liver lobe anteriorly and the pancreas posteriorly. The longitudinal (D1) and anteroposterior diameters (D2) were measured from serosa to serosa between contractions of the gastric antrum. The antral cross-sectional area (CSA) in the RLD position was calculated using the following formula: CSA-RLD (cm^2^) = (D1 × D2 × π) ÷ 4 [11]. The gastric volume was estimated using the following validated formula: gastric volume (mL) in the RLD position = 27.0 + (14.6 × CSA-RLD [cm^2^]) − (1.28 × age) [12].

### 2.3. Anesthesia

Patients entered the operating theater without premedication. After completion of the baseline ultrasound gastric assessment, standard monitoring was initiated, including electrocardiography, non-invasive blood pressure monitoring, pulse oximetry, bispectral index monitoring (Covidien LLC, Mansfield, MA, USA), and neuromuscular monitoring (TetraGraph, Senzime, Uppsala, Sweden).

During spontaneous ventilation with the mouth closed, pre-oxygenation was provided for 3 min by HFNO at 30 L/min using the Optiflow™ system (Fisher & Paykel Healthcare, Auckland, New Zealand). Subsequently, anesthesia induction was performed by propofol (target concentration, 5.0–6.0 μg/mL) and remifentanil (3.0–4.0 ng/mL) using a target-controlled infusion device (Orchestra, Fresenius Kabi, Bad Homburg, Germany). After patients lost consciousness, rocuronium was administered at 0.6 mg/kg, and the oxygen flow rate was increased to 70 L/min while applying jaw thrust with the mouth closed. At 2 min after rocuronium administration, the Cormack–Lehane grade was evaluated using a direct laryngoscope, and a suspension laryngoscope was inserted without tracheal intubation with the HFNO flow rate maintained at 70 L/min with the mouth open. Anesthetic depth was maintained at a bispectral index of 40–60, and propofol and remifentanil were infused at target concentrations of 2.5–4.0 μg/mL and 2.5–5.0 ng/mL, respectively. The heart rate and blood pressure were maintained within 10–20% of the baseline value. The lowest oxygen saturation level during surgery was recorded.

For patient safety, bag valve mask (BVM) ventilation or tracheal intubation was performed after HFNO was discontinued in cases of surgery duration of over 40 min, intraoperative hypoxemia (SpO_2_ < 90%, not relieved by correction of iatrogenic causes), or malignant cardiac arrhythmia during surgery [6].

At the end of surgery, patients were placed in the RLD position for ultrasound gastric assessment. Following the assessment, they were returned in the supine position, and the duration of apnea was recorded just before starting BVM ventilation. The duration of apnea was defined as the time from cessation of spontaneous respiration after administration of rocuronium to the start of BVM ventilation. Starting BVM ventilation, the anesthetic infusions were discontinued simultaneously. BVM ventilation was performed by the one-hand E-C technique using a triple airway maneuver after insertion of an oropharyngeal airway (fresh gas flow 5 L/min, 100% O_2_, tidal volume 8–10 mL/kg, respiratory rate 12 breaths/min, inspiration:expiration ratio 1:2, adjustable pressure-limiting valve 20 cmH_2_O). The highest end-tidal CO_2_ was recorded during the first 30 s of BVM ventilation. For neuromuscular blockade reversal, sugammadex was administered at 2 mg/kg after confirming the second twitch of train-of-four (TOF). Patients opened their eyes voluntarily or upon verbal command, and when proper tidal volume and respiratory rate were restored, the oropharyngeal airway was removed and 100% oxygen at 5 L/min was supplied through a face mask. When the patients’ consciousness and respiration were restored to an appropriate level, they were transferred to the post-anesthetic care unit.

### 2.4. Sample Size Calculation and Statistical Analysis

The full stomach was defined as 1.5 mL/kg of gastric fluid volume. As a previous study reported that the gastric volume of fasting adult patients was 0.7 ± 0.6 mL/kg [13], we defined full stomach more conservatively, as 1.0 mL/kg of gastric fluid volume, because there was no definitive airway that could prevent aspiration in this study population. Since a 50% increase from 0.7 mL/kg is approximately 1 mL/kg, assuming that an increase in gastric volume of 50% or more after HFNO may be clinically significant (α = 0.05 and power (1 − β) = 0.9), and considering possible dropouts, the required number of patients was 45.

Statistical analysis was conducted using IBM SPSS Statistics (version 25.0; IBM Corp., Armonk, NY, USA). Data were presented as means with standard deviations or medians with interquartile ranges, as appropriate. Since measurements were repeated in the same participants, they were analyzed using the paired t-test. Statistical significance was set at *p* < 0.05.

## 3. Results

A total of 45 patients were enrolled in the study. One patient was excluded due to application of tracheal intubation during surgery at the request of the surgeon due to LASER (Light Amplification by the Stimulated Emission of Radiation) use. Therefore, 44 patients were included in the analysis (Figure 1).

The patients’ characteristics are presented in Table 1. The median age was 46 (range, 21–65) years and the mean BMI was 24.4 ± 3.7 kg/m^2^. There were no cases of difficult airway with a Cormack–Lehane grade 3 or higher.

The antral CSA-RLD after HFNO was not significantly different from that at baseline (5.4 ± 1.7 vs. 5.1 ± 1.7 cm^2^, *p* = 0.092), and the difference was −0.3 (95% confidence interval, −0.7 to 0.1) cm^2^. Similarly, the gastric volume (48.1 ± 25.0 vs. 43.8 ± 26.3 mL, *p* = 0.092) and gastric volume adjusted by weight (0.7 ± 0.4 vs. 0.7 ± 0.4 mL/kg, *p* = 0.1) did not show a significant difference between baseline and after HFNO (Table 2).

Table 3 shows the intraoperative oxygenation profiles. The median duration of apnea was 15 (14–22) min. There was only one case of oxygen saturation decrease to less than 90%, but it was recovered within 1 min by reposition of the suspension laryngoscope and jaw thrust. The median value for the highest end-tidal CO_2_ during the first 30 s of BVM ventilation was 58 (54–66) mmHg.

## 4. Discussion

The current study showed that HFNO at 70 L/min for 15 min (up to 35 min) did not increase the gastric volume in patients under tubeless general anesthesia with neuromuscular blockade. In addition, the gastric volume did not increase even when HFNO was administered at 30 L/min for approximately 3 min before rocuronium administration under spontaneous breathing.

Our findings are consistent with those of previous studies. Sud et al. [14] found that when HFNO was used prior to intubation, there was no significant difference in gastric gas volume measured using computed tomography compared to that during facemask ventilation. In their study, HFNO was applied for pre-oxygenation at 30–40 L/min for approximately 5 min during the pre-intubation period, and at 70 L/min for 3–5 min during the apneic period with open mouth jaw thrust [14]. Another study showed that application of HFNO for rapid sequence induction (40 L/min for 3 min for pre-oxygenation and 70 L/min for 2 min during apnea) did not provoke any signs of gastric regurgitation through visual inspection of the pharynx [8]. McLellan et al. [9] reported that HFNO did not increase the gastric volume when the oxygen flow rate was gradually increased from 30 L/min to 70 L/min for 30 min under spontaneous breathing with the mouth closed. Our study differs from these previous studies in that we examined the change in gastric volume after application of HFNO at 70 L/min for 15 min (maximum 35 min) during apnea with the mouth open under total intravenous anesthesia.

HFNO produces CPAP, which is affected by the flow rate, mouth position (open or closed), and breathing status (spontaneous breathing or apnea). The mean nasopharyngeal airway pressure was found to increase by 0.69 cmH_2_O per 10 L/min of HFNO in the closed mouth position and by 0.35 cmH_2_O per 10 L/min of HFNO in the open mouth position under spontaneous breathing [15]. During apnea, the airway pressure increased non-linearly due to flow rate increase, and it remained below 10 cmH_2_O until 80 L/min of HFNO was applied with the mouth closed. During apnea with the mouth open, there was no clinically relevant increase in airway pressure (0.01 cmH_2_O per 10 L/min) [16]. Based on the above, and considering that a PAP greater than 14 cmH_2_O was required to cause gastric distension during facemask ventilation in anesthetized patients [17], the HFNO flow rate of 70 L/min used in our study would not generate sufficient airway pressure to provoke gastric distension irrespective of the mouth position and breathing status.

According to Fitz-Clarke’s predictive model [18], gastric inflation occurs whenever the mouth pressure exceeds the opening pressure of the lower esophageal sphincter. Therefore, the distribution of inhaled volume among the lungs, esophagus, and stomach in an unprotected airway is affected by several variables, such as the lower esophageal sphincter pressure, mouth pressure, airway resistance, and respiratory system compliance [19]. A prior study reported that neuromuscular blockade might reduce the upper esophageal sphincter tone [20], but there is no evidence that it affects the lower esophageal sphincter tone. In our study, HFNO was administered at 70 L/min during apnea with neuromuscular blockade, but the airway pressure likely remained below the opening pressure of the upper esophageal sphincter because it was administered with the mouth open [21].

Gastric insufflation was reported in methods of ventilation with sealing the nose and mouth, such as non-invasive positive pressure ventilation and BVM ventilation. The CPAP used to treat obstructive sleep apnea caused aerophagia in 52% of patients [22]. In a prior prospective study, non-invasive positive pressure ventilation induced aerophagia in 12% of patients with chronic respiratory failure [23]. There was also a reported case of stomach distension caused by bilevel nasal PAP [7]. In BVM ventilation, Ruben et al. [24] reported that pressures < 15 cmH_2_O rarely produced gastric insufflation, while pressures ≥ 25 cmH_2_O resulted in gastric insufflation in most subjects.

In our study, during tubeless anesthesia for laryngeal microsurgery (median apnea period of 15 min), HFNO could maintain proper oxygen saturation, with a low incidence of desaturation. The median lowest SpO_2_ was 98.5%, and there was only one case where the SpO_2_ dropped below 90%. The incidence of SpO_2_ < 95% was 11% (five patients). These findings are consistent with those reported in a recent study by Min et al. [25], which showed that HFNO was not inferior to tracheal intubation in terms of maintaining oxygen saturation during laryngeal microsurgery. In their study, the median lowest SpO_2_ was 100%, and the incidence of SpO_2_ < 90 and SpO_2_ < 95% was 5% and 11%, respectively [25].

There may be concerns of CO_2_ accumulation when using HFNO during apnea. A recent study showed that prolonged apneic oxygenation with HFNO has a limitation in its use as it can be accompanied by severe respiratory acidosis due to CO_2_ accumulation [26]. Although we did not measure blood CO_2_ levels by blood gas analysis, the CO_2_ accumulation levels in our study were deemed acceptable considering that the median highest end-tidal CO_2_ was 58 mmHg during the first 30 s of BVM ventilation. Moreover, this high end-tidal CO_2_ was rapidly reversed and normalized by BVM ventilation. However, caution should be exercised in generalizing our results, which were from end-tidal CO_2_ only. End-tidal CO_2_ value in this study could be affected by many factors; skill of mask fitting, patency of airway, systemic perfusion, pH, dead space, minute ventilation, etc. Although the expert anesthesiologist performed mask fitting and bag-valve mask ventilation in this study, there may still be cases which the difference between the end-tidal CO_2_ and PaCO_2_ increased.

The current study had a few limitations. First, since we applied apneic oxygenation only during brief surgery, our results cannot be extrapolated to longer procedures of several hours. Therefore, further studies including long-term procedures may be necessary. Second, the patient’s mouth was open throughout apneic period due to suspension laryngoscopy. Although airway pressure remained below 10 cmH_2_O up to 80 L/min of HFNO application with closed mouth in the apneic patients, further study may be needed whether or not HFNO cause gastric distension in the closed mouth situation. Third, since patients with a high risk of aspiration and difficult airway were excluded, our results cannot be applied to these patients. Furthermore, there was the possibility of bias, since the sonographer had information about baseline examination while conducting the subsequent examination, and also had knowledge of the hypothesis of the study and the intervention. Lastly, the study was conducted in a single center and with the relatively small sample size; thus, a larger-scale multicenter study is needed to generalize the result of this study.

In conclusion, HFNO at 70 L/min for 15 min (up to 35 min) did not increase the gastric volume in patients undergoing laryngeal microsurgery under tubeless general anesthesia with neuromuscular blockade.

## Figures and Tables

**Figure 1 jcm-12-01800-f001:**
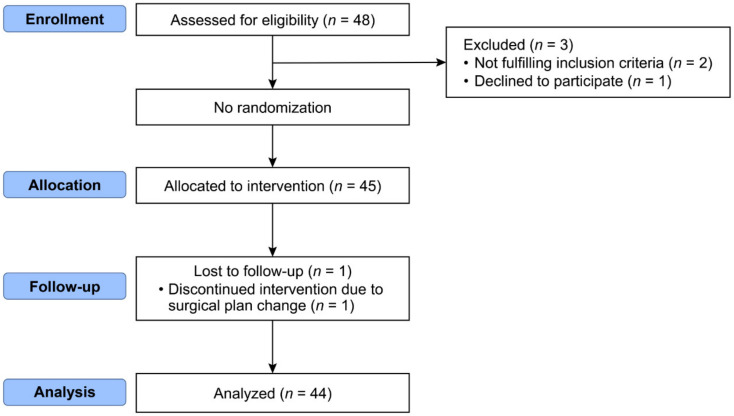
Consort flow diagram.

**Table 1 jcm-12-01800-t001:** Patients’ Characteristics.

	*n* = 44
Age (years)	46.0 (21–65)
Sex (male/female)	24/20
Height (cm)	166.0 ± 9.0
Weight (kg)	68.0 ± 14.0
BMI (kg/m^2^)	24.4 ± 3.7
ASA physical status (I/II)	28/16
Mallampati score (I/II)	12/32
Cormack–Lehane grade (I/IIa/IIb)	26/14/4
Diagnosis	
Polyp	30
Nodule	5
Cyst	3
Neoplasm	2
Reinke’s edema	2
Leukoplakia	2

Values are presented as mean ± standard deviation, median (range), or number. BMI, body mass index; ASA, American Society of Anesthesiologists.

**Table 2 jcm-12-01800-t002:** Antral Cross-Sectional Area and Gastric Volume.

	Baseline	After HFNO	Difference (95% CI)	*p*-Value
Antral CSA-RLD (cm^2^)	5.4 ± 1.7	5.1 ± 1.7	−0.3 (−0.7 to 0.1)	0.092
Gastric volume (mL)	48.1 ± 25.0	43.8 ± 26.3	−4.4 (−9.5 to 0.8)	0.092
Gastric volume (mL/kg)	0.7 ± 0.4	0.7 ± 0.4	−0.1 (−0.1 to 0)	0.1

Values are presented as mean ± standard deviation. HFNO, high-flow nasal oxygenation; CI, confidence interval; CSA-RLD, cross sectional area of the gastric antrum in the right lateral decubitus position.

**Table 3 jcm-12-01800-t003:** Intraoperative Oxygenation Profiles.

	*n* = 44
Duration of operation (min)	10 (5–14)
Duration of anesthesia (min)	35 (30–40)
Duration of apnea (min)	15 (14–22)
Lowest SpO_2_ (%)	98 (97–100)
Incidence of SpO_2_ < 90%	1 (2)
Incidence of SpO_2_ < 95%	5 (11)
Highest end-tidal CO_2_ (mmHg) *	58 (54–66)

Values are presented as median (interquartile range) or number (%). * Highest end-tidal CO_2_ was measured during the first 30 s of bag valve mask ventilation.

## Data Availability

The datasets used and/or analyzed during the current study are available from the corresponding author on reasonable request.

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
