# Peer review of "Effect of High-Flow Nasal Oxygenation on Gastric Insufflation in Patients Undergoing Laryngeal Microsurgery under Tubeless General Anesthesia with Neuromuscular Blockade"

_jcm, 2023, doi:10.3390/jcm12051800_

Round 1

Reviewer 1 Report

Paper analyzing the effect of HFNC on gastric volume change in adult patients undergoing laryngeal microsurgery under tubeless general anesthesia with neuromuscular blockade.

Singles center and with a relative small sample size. The authors must underline the limitations in the generalization of the results.

Why patients ASA higher than 2 were not included?

Please try to provide more information about basal respiratory comorbidities of the patients, as they can influence the results.

Discuss more about the lack of blood gas analysis data during the procedure, as key parameters like PaO2, pH or PaCO2 could be more appropriate to assess the safety of the procedure.

Author Response

Paper analyzing the effect of HFNC on gastric volume change in adult patients undergoing laryngeal microsurgery under tubeless general anesthesia with neuromuscular blockade.

  1. Singles center and with a relative small sample size. The authors must underline the limitations in the generalization of the results.

Thank you for your comment. We added these comments in the discussion section as a limitation.

  1. Why patients ASA higher than 2 were not included?

There are some reasons why we excluded patients with ASA 3 or more.

First reason is related to the patient safety. In this study, about 15 minutes of apnea time (up to 35 minutes) was inevitably occurred, although HFNO was applied during that time. Previous studies which included the patients with ASA 3 or 4 have concluded that an apnea time of 15 minutes with HFNC is safe. However, Benninger et al showed in their case series, six of 53 patients required supplementary methods to supply oxygen due to sustained desaturation (Laryngoscope 2021 Mar;131(3):587-591). One of the randomized controlled trials reported that HFNO was associated with lower intraoperative oxygenation compared to conventional method (SpO2 93.0 ± 5.6% vs. 98.7 ± 1.6%) (Laryngoscope 2020 Dec;130(12):E874-E881). Since the main focus of this study was on the gastric insufflation not safety of HFNO, we selected the patient population more conservatively. 

Second, since this study was conducted with a relatively small sample size, we needed to increase the homogeneity of study population as possible.

  1. Please try to provide more information about basal respiratory comorbidities of the patients, as they can influence the results.

I appreciate your comment. We excluded the patients with any pulmonary disease, but this information was not sufficiently provided in the manuscript. As you pointed out, we added these comments about respiratory comorbidities in the methods section.

  1. Discuss more about the lack of blood gas analysis data during the procedure, as key parameters like PaO2, pH or PaCO2 could be more appropriate to assess the safety of the procedure.

Thank you for the comments and I agree with your opinion. End-tidal CO2 value in non-intubated patients is affected by many factors; skill of mask fitting, patency of airway, systemic perfusion, dead space, minute ventilation, etc. Although the expert anesthesiologist performed mask fitting and bag-valve mask ventilation in this study, still there may be cases which the difference between the end-tidal CO2 and PaCO2 increased. As you pointed out, we added these comments and revised the discussion section.

Reviewer 2 Report

The authors investigated the effect of high-flow nasal oxygenation on gastric volume change in adult patients undergoing laryngeal microsurgery under tubeless general anesthesia with neuromuscular blockade, and found that high-flow nasal oxygenation at 70 L/min during apnea with the mouth open did not influence the gastric volume in patients undergoing laryngeal microsurgery under tubeless general anesthesia with neuromuscular blockade. The present is clear. A few minor concerns that needto be addressed further:

1, The authors stated that “Assuming that an increase in gastric volume of 50% or more after HFNO may be clinically significant”,in page 3. Please explain why is 50%?

2, The duration of the operation were 5-14min, sugammadex 2 mg/kg were used to reverse. Was that enough?

Author Response

The authors investigated the effect of high-flow nasal oxygenation on gastric volume change in adult patients undergoing laryngeal microsurgery under tubeless general anesthesia with neuromuscular blockade, and found that high-flow nasal oxygenation at 70 L/min during apnea with the mouth open did not influence the gastric volume in patients undergoing laryngeal microsurgery under tubeless general anesthesia with neuromuscular blockade. The present is clear. A few minor concerns that need to be addressed further:

  1. The authors stated that “Assuming that an increase in gastric volume of 50% or more after HFNO may be clinically significant”,in page 3. Please explain why is 50%?

The threshold of gastric volume that increases aspiration risk under general anesthesia is not yet well defined. Previous studies defined gastric volume greater than 1.5 ml/kg as high risk of aspiration (Br J Anaesth, 116 (2016), pp. 7-11 and Br J Anaesth 2020 Jul;125(1):e75-e80).   

The gastric volume of fasting adult patients was 0.7 ± 0.6 ml/kg (Br J Anaesth 2019; 122: 79-85). We defined full stomach more conservatively, as 1.0 ml/kg of gastric fluid volume, because there is no definitive airway that can prevent aspiration in this study population. Therefore, an increase of approximately 50% was defined as clinically significant. This was also referred to in previous study (Br J Anaesth 2019 Jan;122(1):79-85).

As you advised, we added these comments and revised the methods section. 

  1. The duration of the operation was 5-14min, sugammadex 2 mg/kg were used to reverse. Was that enough?

Thanks for your comment. We routinely used the neuromuscular monitoring device. When the second twitch of TOF appeared, we administered 2 mg/kg of sugammadex. There may be some misunderstanding, but we did not fix the sugammadex dose on purpose. We have only described our routine clinical practice of laryngeal surgery. Fortunately, there was no case in which sugammadex administration was delayed due to delayed appearance of TOF twitch in this study. Even if the duration of operation was 5 minutes, the actual anesthesia time was about 30 minutes. Therefore, at least 20-25 minutes might have elapsed after rocuronium was administered.

The information was not sufficiently provided in the manuscript, so we added these comments of TOF and revised the methods section.